# Locally Advanced Cervical Cancer in a Patient with Epidermolysis Bullosa Treated with Concurrent Chemoradiotherapy and Electronic Brachytherapy

**DOI:** 10.3390/reports8010012

**Published:** 2025-01-21

**Authors:** Desislava Hitova-Topkarova, Virginia Payakova, Angel Yordanov, Desislava Kostova-Lefterova, Elitsa Encheva

**Affiliations:** 1Scientific and Innovative Program Med for Health, Medical University Pleven, 5800 Pleven, Bulgaria; vpayakova@gmail.com (V.P.); dessi.zvkl@gmail.com (D.K.-L.); 2Department of Radiotherapy, UMHAT “Dr. Georgi Stranski”, 5800 Pleven, Bulgaria; 3Department of Gynecological Oncology, Medical Universiy Pleven, 5800 Pleven, Bulgaria; angel.jordanov@gmail.com; 4National Cardiology Hospital, 65 Konyovitsa Street, 1309 Sofia, Bulgaria; 5Department of Radiotherapy, UMHAT “Saint Marina”, 1, Hristo Smirnenski Blvd., 9010 Varna, Bulgaria; dr.encheva@gmail.com; 6Faculty of Medicine, Medical University Varna, 9002 Varna, Bulgaria

**Keywords:** cervical cancer, radiotherapy, electronic brachytherapy, epidermolysis bullosa

## Abstract

**Background and Clinical Significance**: The purpose of this report is to investigate the feasibility of combined modality treatment in a case of locally advanced cervical cancer in a patient with inherited epidermolysis bullosa as well as to suggest a protocol for cervical electronic brachytherapy. **Case Description**: The patient was treated with image-guided external beam radiotherapy and concomitant chemotherapy to a dose of 45 Gy in 25 fractions with a simultaneously integrated boost of 55 Gy in involved lymph nodes. The maximal skin dose was 34.09 Gy. Intracavitary electronic brachytherapy was applied to the uterine cervix in 4 fractions of 7 Gy and contributed no dose to the skin. **Discussion**: The treatment was tolerated well with no early toxicity. During the 3-month period of follow-up, no adverse events of grade 2 or higher were detected, and no exacerbation of skin lesions was noted. **Conclusions**: This is the first report of treatment of cervical cancer in a patient with inherited epidermolysis bullosa where combined concurrent chemoradiotherapy and intracavitary electronic brachytherapy demonstrated feasibility and safety. The followed institutional protocol for treatment planning and delivery ensured low doses to organs and risk and reproducibility.

## 1. Introduction

Inherited epidermolysis bullosa (EB) is a heterogeneous group of genetic diseases characterized by skin blistering due to minimal trauma [1]. Dystrophic epidermolysis bullosa (DEB) is a type of EB caused by mutations in the COL7A1 gene responsible for the production of type VII collagen.

Prevalence data are scarce, but for EB, it is estimated to be 5:100,000 live births and 2.4:100,000 population in the EU [2]. The prevalence of DEB is 3.3 per 1,000,000 live births in the USA [3]. In Europe, few countries have recent epidemiological data on DEB, and prevalence per 1,000,000 varies from 1.3 to 20.4 [4]. The data from Bulgaria is presented in a single national study from 2001, which states that prevalence rates are 8.6 per 1,000,000 for EB and 3.1 per 1,000,000 for DEB [5].

EB is associated with an increased risk of malignancies at sites of chronic inflammation, i.e., oral cavity cancer [6,7] and skin squamous cell cancer (sSCC) [8,9,10] that have been linked to various pathogenic mechanisms [11] but not to gynecological cancer.

Carcinoma of the uterine cervix is the fourth most common cancer in females, and 80% of cases are squamous cell carcinomas [12]. Staging evaluation is recommended for all patients before initiation of treatment and should include magnetic resonance tomography (MRT) and positron emission tomography/computed tomography (PET/CT). In cases of locally advanced cervical cancer, patients are referred for definitive chemoradiation. The recommended radiotherapy techniques are external beam image-guided radiotherapy (IGRT) followed by image-guided adaptive brachytherapy, and Cisplatin is the chemotherapeutic agent of choice [13,14].

A rare case of locally advanced cervical carcinoma in a patient with inherited EB is presented, which was successfully treated with definitive chemoradiation without additional worsening of the skin lesions.

## 2. Case Description

A 54-year-old woman presented in a gynecology department complaining of postmenopausal bleeding. She had been in amenorrhea for 2 years with no history of previous gynecological illness, one normal childbirth, and no family history of malignancies. She suffers from inherited recessive DEB, which was diagnosed at the age of 32. A biopsy of the uterine cervix revealed invasive squamous carcinoma G2. The patient was referred to the gynecological oncology clinic. Examination confirmed a cervical tumor with a size above 4 cm and no signs of parametrial invasion. MRT and PET/CT findings proved a tumor with an axial size of 46/32 mm and 34 mm craniocaudally with invasion of the vaginal vaults (Figure 1) and multiple bilateral pelvic lymph nodes involved, which were hypermetabolic with a highest SUV max of 42.6 (Figure 2). The stage of the disease was defined as cT2a2 cN1 M0, FIGO IIA. At the tumor board discussion, based on the current guidelines, definitive combined chemoradiotherapy was recommended. The patient presented herself at the radiation oncology department with complaints of intermittent pelvic pain, urinary stress incontinence, and occasional very mild vaginal bleeding. Dystrophic skin lesions were observed on the forearms, calves, and pubic area. Severe dystrophy of the nails was present. Recorded risk factors included obesity (body mass index 36.7) and tobacco smoking.

The patient was simulated in a prone position with hands above the head and was immobilized with a thermoplastic mask covering the abdominal and pelvic area. Knee and feet rests were also used for immobilization. Treatment planning computed tomography (CT) protocol included oral administration of 50 mL iodine contrast media in 500 mL of water 30 min before simulation and CT images with 2 mm slice thickness. The treatment planning system (TPS) Monaco 5.1. was used for dosimetric calculation. The contoured organs at risk (OAR) included the small bowel, liver, spleen, pancreas, stomach, spinal cord, kidneys, bladder, rectum, sigmoid colon, skin, and femoral heads and necks. The clinical target volume (CTV) included the uterine body, cervix and parametria, the upper vaginal third, and regional lymph nodes: common, internal, and external iliac, obturator, presacral, and paraaortic up to the renal vessels. The prescribed dose to the planned target volume (PTV) was 45 Gy in 25 fractions, five times a week, with a CTV to PTV margin of 5 mm. The involved lymph nodes received a simultaneous integrated boost dose of 55 Gy in 25 fractions with a 3 mm safety margin from the gross tumor volume to the PTV. Dosimetric planning was performed using a single 360–arc with 6 MV photons. The achieved target coverage was as follows: 95% of the prescribed dose covered more than 95% of the PTVs, and 100% of the prescribed dose covered more than 99% of the CTVs. OARs met both the soft and hard constraints from the EMBRACE II protocol [15] except for the bowel and sigmoid colon, where only the soft constraints were achieved. In this particular case, additional actions were necessary during dosimetric planning to ensure the skin’s minimal radiation exposure. To evaluate skin dose, a structure covering the outer 5 mm of the body surface was created. The maximal skin dose was 34.09 Gy, with 30 Gy in 1.06 cm^3^, 20 Gy in 53.32 cm^3^, and 10 Gy in 552.84 cm^3^ (11.8% of the structure’s total volume). The maximal dose to the pubic skin was 12 Gy. The total and skin dose distributions are represented in Figure 3.

After pre-treatment quality assurance, the patient began the external beam IGRT and concurrent chemotherapy with Cisplatin 40 mg/kg weekly. The first Cisplatin infusion was given with the first fraction, and the patient received a total number of five cycles. External beam radiotherapy (EBRT) was performed on a linear accelerator Elekta Synergy Platform with daily cone beam CT imaging ensuring the proper patient set-up and reproducible bladder and rectum filling. The EBRT was delivered in 38 days. The treatment was well tolerated by the patient, and no early toxicity was detected during the chemoradiation course. She was scheduled for an MRT scan two days before the last EBRT fraction to assess the tumor response, depth of uterine cavity, and preferred angle of the tandem for cervical brachytherapy. The MRT scan revealed that the gross tumor had shrunk to 15 mm in its maximal diameter, and all previously enlarged lymph nodes were undetectable except for the right external iliac lymph node, the size of which decreased from 29 mm to 15 mm with a necrotic center (Figure 4). After gynecological examination and signing an informed consent, the patient was scheduled for intracavitary electronic brachytherapy (EBT).

A mobile Xoft Axxent electronic brachytherapy system was used with sets of cervical applicators comprising tandems and ovoids. The radiation source is a miniature X-ray tube with 50 KV energy. Minimal shielding is required during the irradiation process, and the participating medical personnel could stay in the room during irradiation behind a screen.

The first fraction of EBT was scheduled for the last day of EBRT to decrease the overall treatment time. The verification that the position of the uterus was appropriate for the 45° tandem was performed by installing a urethral catheter with 200 mL saline infusion and acquiring a cone beam CT image. The patient was positioned lying on a vacuum bag on the operating table and was given short-acting intravenous anesthesia. The insertion of the applicator was performed by a gynecologic oncologist under transabdominal ultrasound guidance. The 45° tandem was inserted to a depth of 4 cm with cervical stopper device and the ovoids were 25 mm in diameter. The applicator was locked to the baseplate by the clamp. When no pain was reported by the patient, she was transported to the CT room for a pelvis CT scan with a 2 mm slice thickness. Images were transferred to the Brachycare TPS, and meanwhile the bladder was emptied for patient’s comfort, followed by patient transfer to the treatment room. Defined OARs were bowel, rectum, bladder, and sigmoid. The target volumes were the CTV high risk covering the whole cervix and CTV intermediate risk, generated with 10 mm margins in lateral and craniocaudal directions and 5 mm in the anteroposterior direction, avoiding the OARs. The prescribed dose per fraction was 7 Gy to the CTV high risk (Figure 5) and the doses in OARs are represented in Table 1.

The plan was transferred to the controller and executed smoothly after filling the bladder with 200 mL of saline again. After completion of the treatment session, the applicator was removed, and the patient spent the rest of the day comfortably.

The second fraction EBT was delivered the next day and two more fractions on consecutive days were given the following week, to a total EBT dose of 28 Gy in four fractions to the uterine cervix. The overall treatment time was 48 days. No immediate toxicity was reported by the patient or by the treating team. The appearance of the EB lesions is presented in Figure 6 and Figure 7. A follow-up MRT scan and gynecological examination were performed after a month.

The first follow-up examination revealed mild fatigue and no exacerbation of EB. The patient reported returning to her job and graded her overall health as 5 on a scale from 1 to 7, which was consistent with the pre-treatment value. No visible cervical tumor was found, and mild vaginal discharge was noted on the gynecological examination. Contrast-enhanced MRT scan also did not detect any cervical tumor, but one right external iliac node was persistent with reduced size from 29 mm to less than 9 mm on LAVA and T2 sequences (Figure 8). Gynecological examination 2 months post-treatment did not observe any evidence of remaining tumor nor early adverse events. The presented urinary incontinence before the treatment persisted without any additional worsening. The patient was scheduled for a PET/CT for a further response evaluation 3 months after treatment, which revealed persistence of the right iliac node with a SUV max of 32.3. The tumor board referred the patient for stereotactic body radiosurgery.

## 3. Discussion

EB is reported to be associated with the development of skin squamous cell carcinoma (sSCC), as the tumors tend to arise at sites of chronic skin trauma. Treatment of sSCC with radiotherapy is recommended in small daily fractions (e.g., 2 Gy) to prevent severe skin toxicity. The use of systemic chemotherapy is generally not recommended and must be applied with caution due to the risk of the development of life-threatening infections according to the guideline published in 2016 [16]. Following literature reviews on sSCC in EB patients explored various treatment options including EGFR inhibitors and immune checkpoint inhibitors. However, they have demonstrated mixed results, and no novel treatments can be generally recommended [8,9,10,17].

There are only a few cases reported in the literature of patients with non-cutaneous malignancies and EB. The published cases of radiotherapy application in such patients are presented in Table 2.

Gynecological cancer is not commonly observed in EB patients. One case of vulvar cancer and EB was found in the literature where radiotherapy was omitted [26] and only one occurrence of cervical carcinoma in a patient with acquired EB, deemed to be paraneoplastic [27].

To our best knowledge, no cases of cervical cancer in patients with inherited EB have been published yet. Based on the present guidelines and reports, it was assumed that chemoradiotherapy was not contraindicated in case the condition of the skin lesions was not complicated by infection or blistering. The patient was treated with modern radiotherapy techniques and with great caution to her skin condition. The necessity to irradiate a large field percutaneously required a balance between skin and bowel doses. Consequently, hard constraints were not met in order to keep the skin volume receiving 30 Gy as low as possible. The patient was strictly monitored for signs of infection or bone marrow toxicity. Low-energy electronic brachytherapy provided the opportunity to deliver a high dose of radiation to the tumor volume with minimal doses to the organs at risk and no further irradiation of the skin. A detailed published algorithm and protocol for cervical EBT is lacking in the literature, whereas the described institutional protocol could serve as a starting point in future studies. The presented clinical and dosimetric data are consistent with a previously published study on the treatment of cervical cancer with EBT [28].

In this case, all necessary treatment modalities were applied without any early toxicity. Skin doses below 35 Gy and the administration of five cycles of Cisplatin did not worsen the DEB lesions. Nevertheless, the data only pertains to a single patient, which is insufficient to draw broad inferences regarding DEB patients’ tolerance to chemoradiotherapy. Moreover, late toxicity of the described treatment could not be appropriately assessed, as a therapy of another modality was consequently delivered. In EB and cancer patients with presentation of skin infection or severe blistering, omission of chemotherapy might be discussed in order to avoid immunosuppression. The establishment of an international registry would give researchers and physicians the chance to learn, explore, and develop novel treatment options, especially given the rarity of the combination of EB and oncological disorders.

## 4. Conclusions

Combined concurrent chemoradiotherapy and intracavitary electronic brachytherapy demonstrated feasibility and safety in the case of locally advanced cervical cancer in a patient with recessive dystrophic epidermolysis bullosa. Further investigation of the subject would be beneficial to clinicians.

## Figures and Tables

**Figure 1 reports-08-00012-f001:**
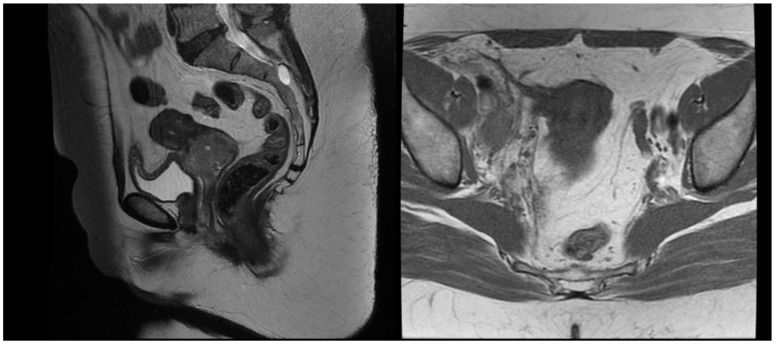
T2 weighted MRT images before treatment.

**Figure 2 reports-08-00012-f002:**
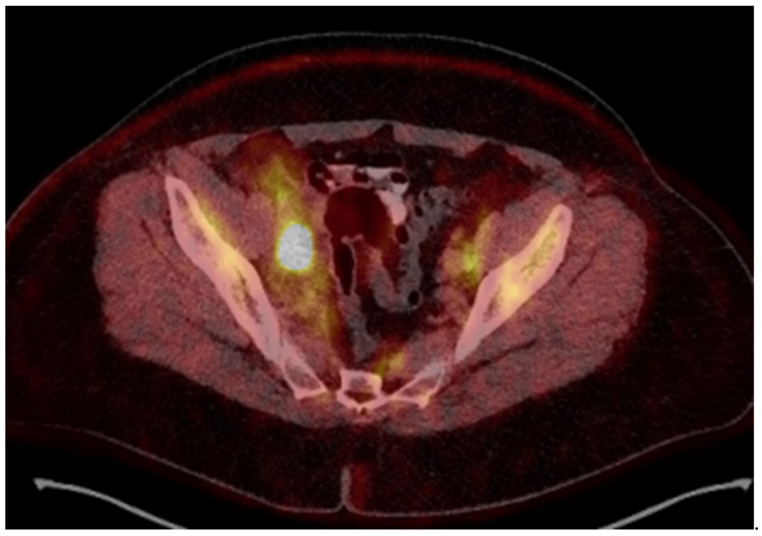
18-Fluorodeoxyglucose (FDG) PET/CT image before treatment.

**Figure 3 reports-08-00012-f003:**
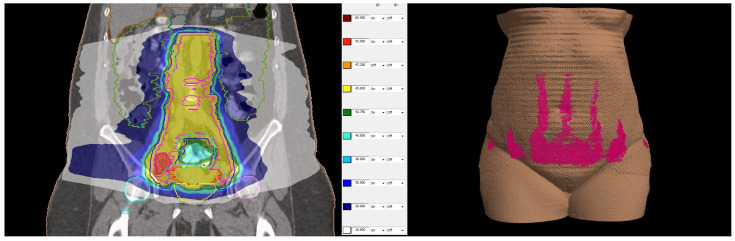
(**Left side**): dose distribution from 55 Gy, which is 100% of the prescribed dose (in red) in boosted lymph nodes and 45 Gy (in yellow) prescribed dose in the whole PTV to 10 Gy (in white). (**Right side**): Three-dimensional reconstruction of the patient’s skin with the distribution of 10 Gy in pink. All regions of existing skin lesions were avoided and received less than 10 Gy.

**Figure 4 reports-08-00012-f004:**
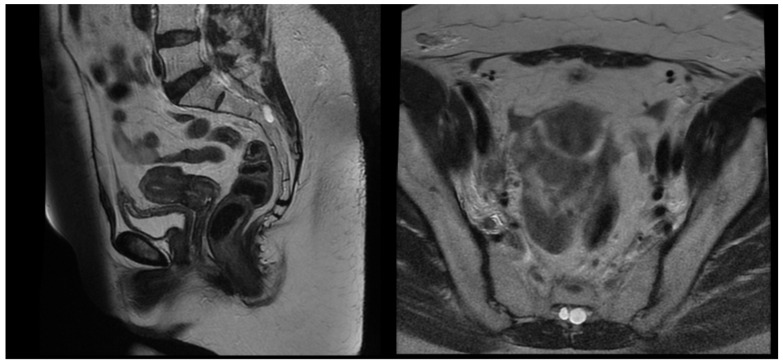
T2 weighted MRT images after EBRT.

**Figure 5 reports-08-00012-f005:**
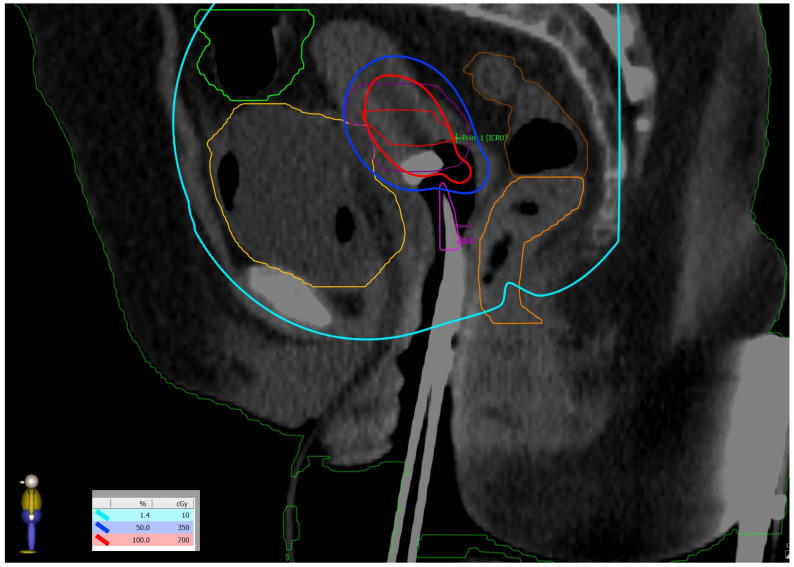
Dose distribution: prescribed dose per fraction—7 Gy (red isoline)—covers more than 90% of the CTV high risk; 50% of the prescribed dose per fraction—3.5 Gy (blue isoline)—covers more than 99% of the CTV intermediate risk. The dose on the skin surface was 0 Gy from EBT.

**Figure 6 reports-08-00012-f006:**
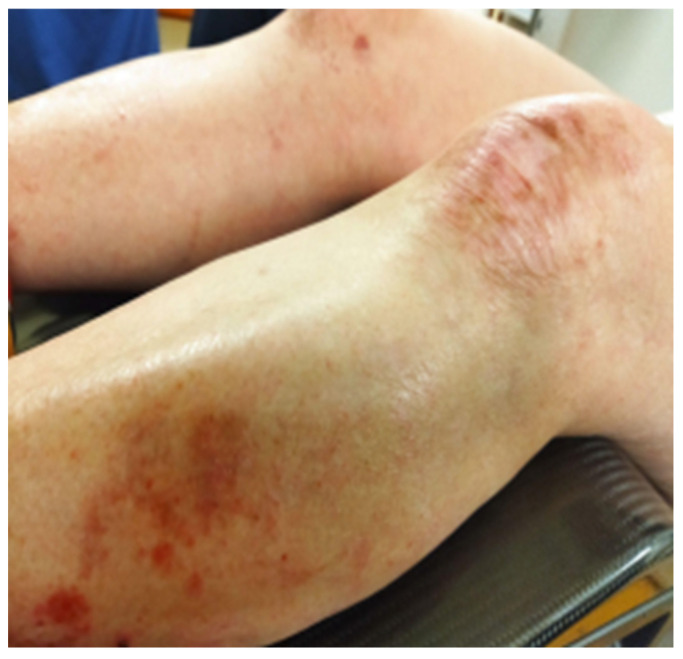
Dystrophic lesions on the lower extremities photographed on the last day of EBRT and after five cycles of Cisplatin.

**Figure 7 reports-08-00012-f007:**
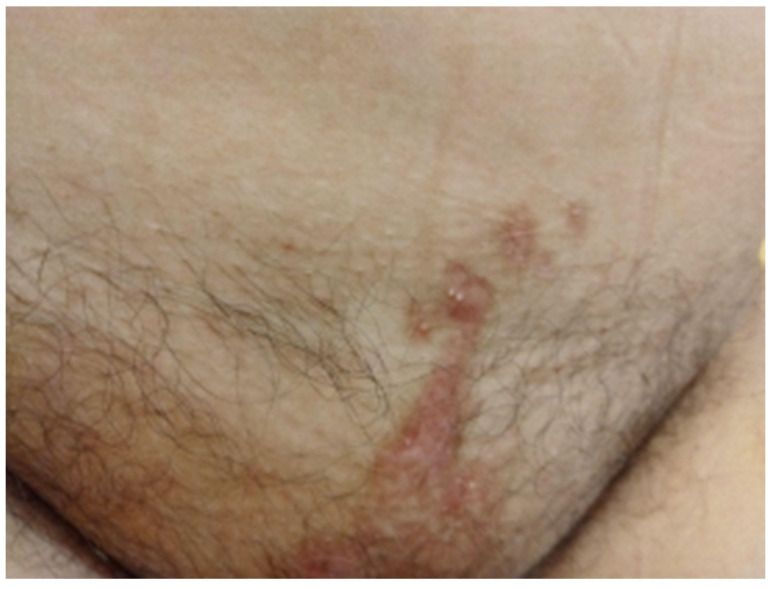
Pubic skin area remained without any signs of infection or blistering after completion of therapy.

**Figure 8 reports-08-00012-f008:**
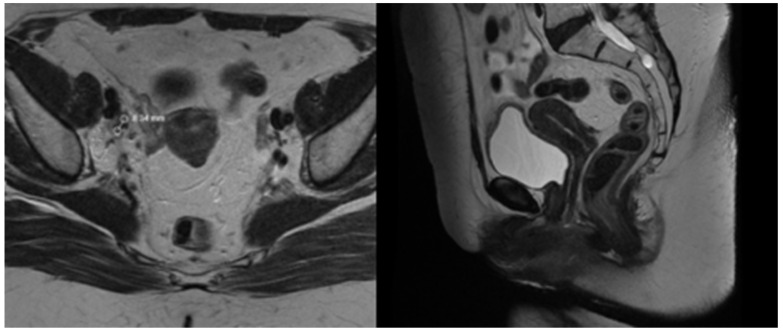
1 month post-treatment MRT images.

**Table 1 reports-08-00012-t001:** Average doses per fraction from EBT. V50% and V35%—the percentages of the OAR receiving 50% or 35% of the prescribed dose; D_2cc_, D_1cc_, and D_0.1cc_—maximum doses to the OAR in 2cc, 1cc, and 0.1 cc of the contoured organ volume.

OARs.					
	V_50%_ (%)	V_35%_ (%)	D_2cc_ (Gy)	D_1cc_ (Gy)	D_0.1cc_ (Gy)
Bladder	0.9	2.8	3.71	4.34	5.96
Rectum	0	0	0.58	0.88	0.99
Sigmoid	0.9	4	2.32	3.66	4.08

**Table 2 reports-08-00012-t002:** Published data on delivered doses to skin or extracutaneous areas and outcomes concerning the skin in patients with EB.

Reference	Radiotherapy Target	Prescribed Dose	Reported Skin Dose	Outcome
Bastin [18]	skin		>45 Gy	moist desquamation, delayed healing
Koulis [19]	pelvic lymph nodes	48 Gy	<40 Gy	grade 1 erythema
Al Shareef [20]	tongue	30 Gy		ulceration, hemorrhage and necrosis)
Ong [21]	parotid gland, ipsilateral lymph nodes levels IB- V	60 Gy, 52 Gy		moist desquamation with complete healing
Bavishi [22]	brain temporal lobe, craniospinal irradiation	32.4 Gy, 23.4 Gy		moist desquamation G2, bullae, hyperpigmentation
Mizutani [23]	larynx	unknown		none reported
AlKhawajah [24]	nasopharynx	70 Gy with concurrent Cisplatin		clearance of blisters
Ray [25]	esophagus	61.2 Gy		Grade 2 erythema and desquamation

## Data Availability

The authors declare that all related data are available from the corresponding author upon reasonable request. The data are not publicly available due to privacy concerns.

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
