# Peer review of "Locally Advanced Cervical Cancer in a Patient with Epidermolysis Bullosa Treated with Concurrent Chemoradiotherapy and Electronic Brachytherapy"

_reports, 2025, doi:10.3390/reports8010012_

Round 1

Reviewer 1 Report

Comments and Suggestions for Authors

This is the first documented case of concurrent chemoradiotherapy and intracavitary electronic brachytherapy for cervical cancer in a patient with inherited epidermolysis bullosa (EB). The study fills a significant gap in the literature. The use of modern radiotherapy techniques with careful adaptation to the patient's unique condition is commendable.Suggerisco di affrontare I seguenti punti per implementare il paper:

- Several sentences in the manuscript are verbose or lack clarity.

- Revise for conciseness and to improve flow in sections like the Introduction and Discussion

- Ensure all acronyms (e.g., EB, IGRT, PET/CT) are clearly defined upon their first mention. While most are defined, some could be overlooked by readers unfamiliar with radiotherapy or EB-specific terminologies.

 - Acknowledge the small sample size (single patient) and the rarity of the condition as key limitations, emphasizing that findings may not be generalizable.

-  Discuss the limitations of the applied protocol, such as whether the methodology would need adaptation for patients with more severe EB or those with comorbid conditions.

- Mention the challenges of assessing long-term toxicity due to the short follow-up period.

- Given the rarity of the condition, propose creating international registries to document similar cases for collective learning.

- Explore in literature alternative chemotherapy regimens tailored to reduce infection risks and systemic side effects for patients with genetic skin fragility disorders.

- The figures provided are informative, but captions should include more details about their relevance to the main findings (e.g., how dose distributions align with safety thresholds for EB patients).

The manuscript is well-prepared and contributes significantly to a rarely documented subject.

Author Response

Dear reviewer,

Thank you very much for taking the time to review this manuscript. Please, find the responses to the comments below, as well as the revised manuscript.

Comments 1: 

  • Several sentences in the manuscript are verbose or lack clarity.
  •  Revise for conciseness and to improve flow in sections like the Introduction and Discussion

Response 1: Thank you for pointing this out. We have revised the text and made multiple corrections.

Comments 2: Ensure all acronyms (e.g., EB, IGRT, PET/CT) are clearly defined upon their first mention. While most are defined, some could be overlooked by readers unfamiliar with radiotherapy or EB-specific terminologies.

Response 2: We are glad that you have noticed and helped us correct the lack of explanation of some abbreviations. Now terms like CT and PTV have been defined.

Comments 3: Acknowledge the small sample size (single patient) and the rarity of the condition as key limitations, emphasizing that findings may not be generalizable

Response 3: In the final revised version, this information is included in lines 231-233.

Comments 4:  Discuss the limitations of the applied protocol, such as whether the methodology would need adaptation for patients with more severe EB or those with comorbid conditions. 

- Mention the challenges of assessing long-term toxicity due to the short follow-up period.

- Given the rarity of the condition, propose creating international registries to document similar cases for collective learning.

Response 4: Thank you very much, the propsed changes have been made in the discussion section.

Comments 5:  Explore in literature alternative chemotherapy regimens tailored to reduce infection risks and systemic side effects for patients with genetic skin fragility disorders

Response 5:  In the final version of the revised manuscript, the chemotherapy options are discussed in lines 194-200.

Comments 6: The figures provided are informative, but captions should include more details about their relevance to the main findings (e.g., how dose distributions align with safety thresholds for EB patients).

Response 6:  We are grrateful for the insight and have added changes to the descriptions of figures 3 and 5.

Reviewer 2 Report

Comments and Suggestions for Authors

Dear Authors, EB is a rare disease, and its association with cancer is even rarer. Having a series on this by one author is difficult, so this case report has its own value. The protocol is understandable.

1) This is a first report of treatment of cervix cancer in a patient with IEB that highlights the feasibility of concurrent chemoradiotherapy and intracavitary brachytherapy
2) This case report will be valuable to someone who has patient with this kind of rare disease and probably later will be  help in writing a review or guidelines. To be of this value , it should have complete methodology and adverse effect/complications. This case report has fulfilled the the first thing but there were no adverse effects after giving standard treatment.
5) Apart from the spell checks and vocabulary, I can suggest that the authors can add a table showing the literature
6) Even though they have observed no adverse effects. They can elaborate on expected radiation induced skin reactions.
7) The best thing would be to add a paragraph on what exactly they did different in this case in comparison to their routine cervix cancer patient 

There are a few typo errors, (1)Line 66 instead of slash use X(2) SUV max 42,6? (3)punctuation in lines 67,110,198 and a few more typos. 

Author Response

Dear reviewer, thank you very much for taking the time to review this manuscript. Your comments were of great value to us. Please find the responses to the comments below and the revised manuscript.

Comments 1: Apart from the spell checks and vocabulary, I can suggest that the authors can add a table showing the literature

Response 1: Thank you for noting this. We have made improvements in the spelling and you can find table 2 in the revised manuscript line 204.

Comments 2: Even though they have observed no adverse effects. They can elaborate on expected radiation induced skin reactions.

Response 2: We are grateful for pointing this out. The expected reactions have now been mentioned in table 2 as well as in line 196 where the guideline is cited.

Comments 3: The best thing would be to add a paragraph on what exactly they did different in this case in comparison to their routine cervix cancer patient

Response 3: We have tried to point out the differences and challanges in a more clear way now in the discussion section, lines 215-224.